



# Comparison of meteor radar and TIDI winds in the Brazilian equatorial region

Ana Roberta Paulino[1], Delis Otildes Rodrigues[1], Igo Paulino[2], Lourivaldo Mota Lima[1], Ricardo Arlen Buriti[2], Paulo Prado Batista[3], Aaron Ridley[4], and Chen Wu[4]

[1]Departamento de Física, Universidade Estadual da Paraíba. Rua Baraúnas, 351. Campina Grande, PB, Brazil.
[2]Unidade Acadêmica de Física, Universidade Federal de Campina Grande. Rua Aprígio Veloso, 882. Campina Grande, PB, Brazil.
[3]Divisão de Heliofísica, Ciências Planetárias e Aeronomia, Instituto Nacional de Pesquisas Espaciais. Avenida dos Astronautas, 1.758. São José dos Campos, SP, Brazil.
[4]University of Michigan, 1416 Space Research Building Ann Arbor, MI 48109-2143, USA.

**Correspondence:** A. R. Paulino (arspaulino@gmail.com)

**Abstract.** Using data collected from a meteor radar deployed at São João do Cariri ($7.4^o$, $36.5^0$S) and the TIMED Doppler Interferometer (TIDI) on board the Thermosphere-Ionosphere-Mesosphere Energetics and Dynamics (TIMED) satellite for 2006, comparisons of the horizontal winds (meridional and zonal components) were made in order to evaluate these techniques for scientific investigation and pointed out advantages of each instrument. A grid of $\pm$ 5 degrees of latitude and longitude

centered at São João do Cariri was used to calculate the mean winds from the TIDI, which have a resolution of 2.5 km altitude starting from 82.5 km up to 102 km altitude. Otherwise, the meteor radar computes the winds for 7 layers of 4 km thickness overlapping 0.5 km above and below, which produces layers spaced by 3 km from 81 to 99 km altitude. When almost simultaneous measurements were compared, substantial discrepancies were observed in the vertical wind profiles. It happened because the meteor radar uses one hour bin size to estimate the wind from the echoes detected in the whole sky. While the

TIDI measures instantaneous winds from the airglow emissions. In contrast, when the longer period of observation was taken into account, the meteor radar daily winds, averaged within a time interval of one month, were smoothed and showed more clearly the characteristics of the propagation of tides. The responses of the horizontal wind to the intraseasonal, semiannual and annual oscillations were satisfactory for the both techniques.

## 1 Introduction

The mesosphere and lower thermosphere (MLT) is rich in dynamical processes. A large spectrum of mechanical oscillations can be observed in this region, which includes acoustic waves, gravity waves, atmospheric tides, planetary waves, seasonal oscillations, quasi-biennial oscillations and so on. Those phenomena are important to understanding the general circulation of the atmosphere because the wave propagation can transfer energy and momentum among different levels of the atmosphere (Smith, 2012).

Wind measurements in the MLT are important to investigate the interaction between the background atmosphere and waves (e.g., Hindley et al., 2022). The main instruments that have been used to estimate the wind in this region are: meteor radar




(e.g., Buriti et al., 2008); mesosphere-stratosphere-troposphere radar (e.g., Balsley et al., 1980; Qiao et al., 2020); middle and upper atmosphere radar (e.g., Fukao et al., 1985); laser imaging, detection, and ranging (LIDAR) (e.g., Clemesha et al., 1981); medium frequency radar (e.g., Igarashi et al., 1996) and Fabry-Perot interferometer (e.g., Fujii et al., 2004). In the last decades,
satellite measurements of the wind have contributed to understanding global responses of planetary, tidal and gravity waves and other using wind measurements (e.g., Killeen et al., 2006; Niciejewski et al., 2006).

The meteor radar is a relatively moderate cost instrument used in the studies of the MLT dynamics. Generally, the meteor radar can estimate hourly horizontal wind from 80 to 100 km altitude (e.g., Paulino et al., 2015). This time sample is very good to investigate long period oscillation like tidal and longer period waves (e.g., Lima et al., 2006). Additionally, those kinds
of measurements have also been used to investigate the background conditions of the atmosphere for the propagation of short period gravity waves (e.g., Fechine et al., 2009; Bageston et al., 2011; Carvalho et al., 2017, and references therein).

On the other hand, the satellite measurements can provide instantaneous winds. The TIMED Doppler Interferometer instrument (TIDI) on board of the Thermosphere-Ionosphere-Mesosphere Energetic Dynamics (TIMED) satellite can provide horizontal winds with 2.5 km vertical resolution from 82.5 km up to 102.5 km (Killeen et al., 2006; Niciejewski et al., 2006).
Besides, the high resolution sample of the TIDI measurements is useful to investigate short period gravity waves (e.g., Baumgarten et al., 2018).

In the attempt of better understanding the potential of the satellite wind measurements, some questions appear: (i) how does TIDI winds compare with meteor radar measurements? (2) what are the advantages and disadvantages of each technique? Some works have been published elsewhere trying to answer such questions (e.g, Xu et al., 2009; John et al., 2011; Su et al.,
2014). The present work aims to advance in this topic comparing measurements of meteor radar deployed at São João do Cariri ($7.4^o$S, $36.5^o$W) to the TIDI measurements for a grid of $5^o \times 5^o$ (latitude $\times$ longitude) centered at São João do Cariri. Salient aspect of instantaneous and long period observation will be presented and discussed.

## 2  Instrumentation and Observations

The meteor radar is a transceiver consisting of an interferometric receiver set of five yagi antennas of two elements, a transmitter
yagi antenna of three elements, a receiver and a transmitter modulus. It operates at 35.24 MHz emitting 2144 pulses per second. The meteor radar uses the ablation of the meteoroids that penetrate in the MLT region. The ionized trails serve to reflect the transmitted radio waves back to ground as meteor echoes. Operating with a power of 12 kW, the meteor radar can detect between 1,000 and 3,000 echoes per day (e.g., Hocking et al., 2001; Paulino et al., 2015).

The travelling time of the radio waves from the transmitter antenna, reflecting in the trail and coming back to the receiver
antennas allows to calculate the distance of the detected meteor. The set of receiver antennas presents in an asymmetric cross configuration is used to estimate the location of the meteor in the sky. Lastly, the Doppler shift of the signal gives the information of the wind that is pushing the meteor trails (e.g., Paulino et al., 2012).



Using the parameters described above, the next step is to estimate the mean wind. It is necessary to define vertical and temporal bin sizes to estimate the northward and eastward winds. In the present work, seven layers of 4 km thickness overlapping

0.5 km above and below were used in the vertical, while the temporal resolution was one hour (Clemesha et al., 2001).

Basically, the TIMED Doppler Interferometer (TIDI) is a Fabry-Perot interferometer on board the Thermosphere-Ionosphere-Mesosphere Energetics and Dynamics (TIMED) satellite. The TIDI is equipped with a charged coupled device (CCD) and has four identical telescopes, besides the modulus of control and operation (Killeen et al., 2006).

The TIDI was designed to measure wind and temperature in the MLT region from 70 to 120 km altitude using the airglow

as tracer. The interferometer measures the radiation from the OI5577 and rotational line of $O_2(0,0)$ airglow emission. It has vertical resolution of 2.5 km and an accuracy of $\sim$3 m/s for the estimated wind (Skinner et al., 2003; Niciejewski et al., 2006).

Thus, in the MLT region there is a vertical measurements overlapping of the meteor radar and TIDI that can be used to comparisons and consequently, it is possible to identify advantages of each instrument for different kinds of scientific investigations. The present work aims to contribute to this topic by comparing data collected during 2006 by a meteor radar

deployed at São João do Cariri (7.4$^o$S, 36.5$^o$W) with the measurements by the TIDI, considering a geographical grid of $\pm$ 5 degrees of latitude and longitude, centered at São João do Cariri.

## 3    Data Analysis and Discussion

Figure 1 shows vertical profiles for meridional (blue) and zonal (red) winds. Solid lines represent the meteor radar measurements at 14:00 universal time (UT) on 15 March 2006, while the dashed (14:13 UT) and dot-dashed (14:17 UT) represent the

TIDI measurements on the same day. One can observe that there are large discrepancies between the two measurements, even within a short time interval.

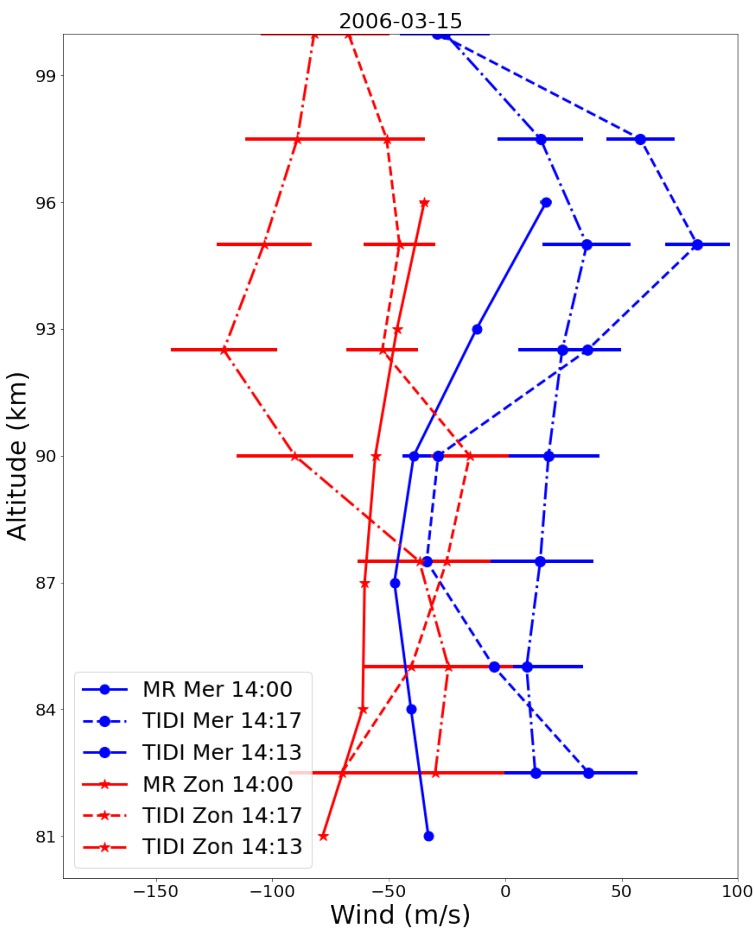

**Figure 1.** Vertical profiles of the meridional (blue) and zonal (red) winds measure by the meteor radar (solid lines) and TIDI (dashed and dot-dashed lines) over São João do Cariri. The meteor radar profiles were calculated at 14:00 UTC on 15 March 2006. The TIDI measurements were retrieved at 14:13 (dashed lines) and 14:17 (dot-dashed lines).

On the one hand, the calculation of the mean winds from the meteor radar uses the bin size of one hour computing all meteors within its field of view. On the other hand, the TIDI estimates the wind from the airglow within $2.5^o$ horizontally during a single sounding (Killeen et al., 2006). Therefore, the wind profiles from the two instruments could be largely different.

Another thing that calls the attention is that the TIDI measurements are quite different, even within a time interval of 4 minutes. It is well known that the wind in the MLT can change quickly (e.g., Clemesha et al., 1981; Kishore Kumar et al.,





2018), primarily, as response to the passage of gravity waves in this region (Baumgarten et al., 2018). However, as showed by John et al. (2011), if a longer interval like 2-3 h is taken into account, the profiles will get close enough.

The results from Figure 1 demonstrate that the satellite measurements are more reliable to investigate the propagation of
gravity waves and their interaction with the background atmosphere. For instance, winds measured by radar have been used to evaluate the background condition of the atmosphere in the creation of Doppler ducts in the MLT (Fechine et al., 2009; Bageston et al., 2011; Carvalho et al., 2017, e.g.,), that are necessary conditions for the propagation of ducted waves in the MLT (Dewan and Picard, 1998). Indeed, the usage of the TIDI wind for case studies of mesosphere fronts could produce more confident results.

Another important contribution for the studies of ducted gravity waves is the interaction of them with the background atmosphere that can produce either convective or dynamic instabilities (Fritts and Rastogi, 1985). The most common parameter used for classifying the instability as convective or dynamic is the Richardson number, which is the ration between the buoyancy and wind shear. Thereby, the TIDI is indeed advantageous.

The real disadvantage for investigating gravity waves, observed by local instruments is that it is necessary to have coincident
crossing of the satellite over the point of observation, which could be not easy to occur.

How are the climatological winds from the TIDI reliable, since they have strong short time variations? In order to try to answer this question, a climatological mean wind was calculated for each month of 2006 for the meteor radar (Figures 2, 4) and TIDI (Figures 3, 5).

Figure 2 shows the daily mean meridional wind calculated using all days within the months as a function of the altitude.
One can observe that the meridional winds range approximately between -120 to 100 m/s, where the large amplitudes were observed in the summer months. It is clear the presence of diurnal oscillation propagating with the decrease of the altitude in most of the months. For some months, primarily in the autumn and winter, the semidiurnal oscillations appear dominant in the high levels. This annual variability of the diurnal and semidiurnal tides is well known in the equatorial region (e.g., Lima et al., 2007). As the wind calculated by the meteor radar data is averaged in a time bin size of one hour, none short time (< 2 h)
oscillations were observed clearly.



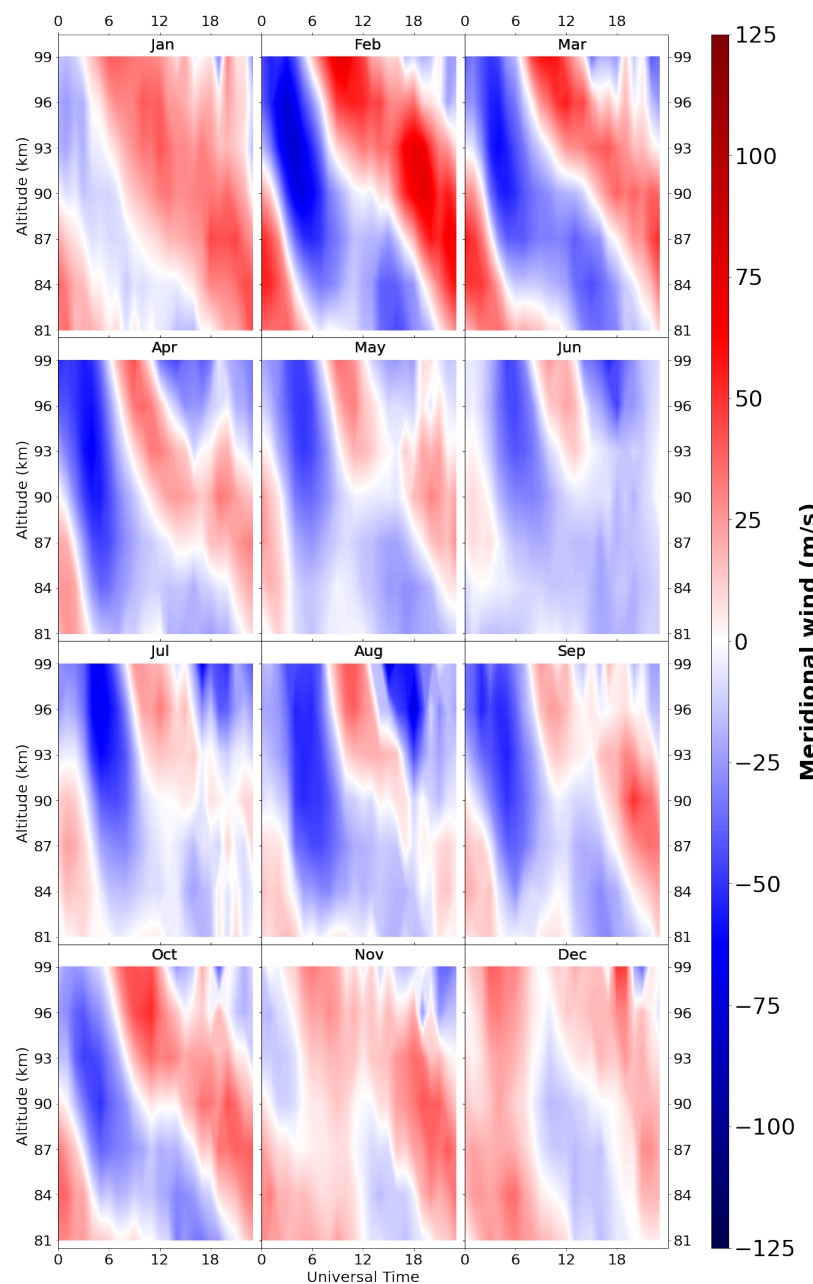

**Figure 2.** Monthly meridional mean wind calculated using the meteor radar for 2006.

Figure 3 is similar to Figure 2, but for the TIDI meridional winds. It has been produced using the data retrieved from the TIDI within a grid of $\sim 10^o$ latitude and longitude centered at São João do Cariri along of 60 days centered in each month of



2006. The color bar is in the same scale of Figure 2, thereby, the amplitudes of the wind are quite similar comparing the two kind of measurements. This good qualitative comparison was concluded by Xu et al. (2009) as well.

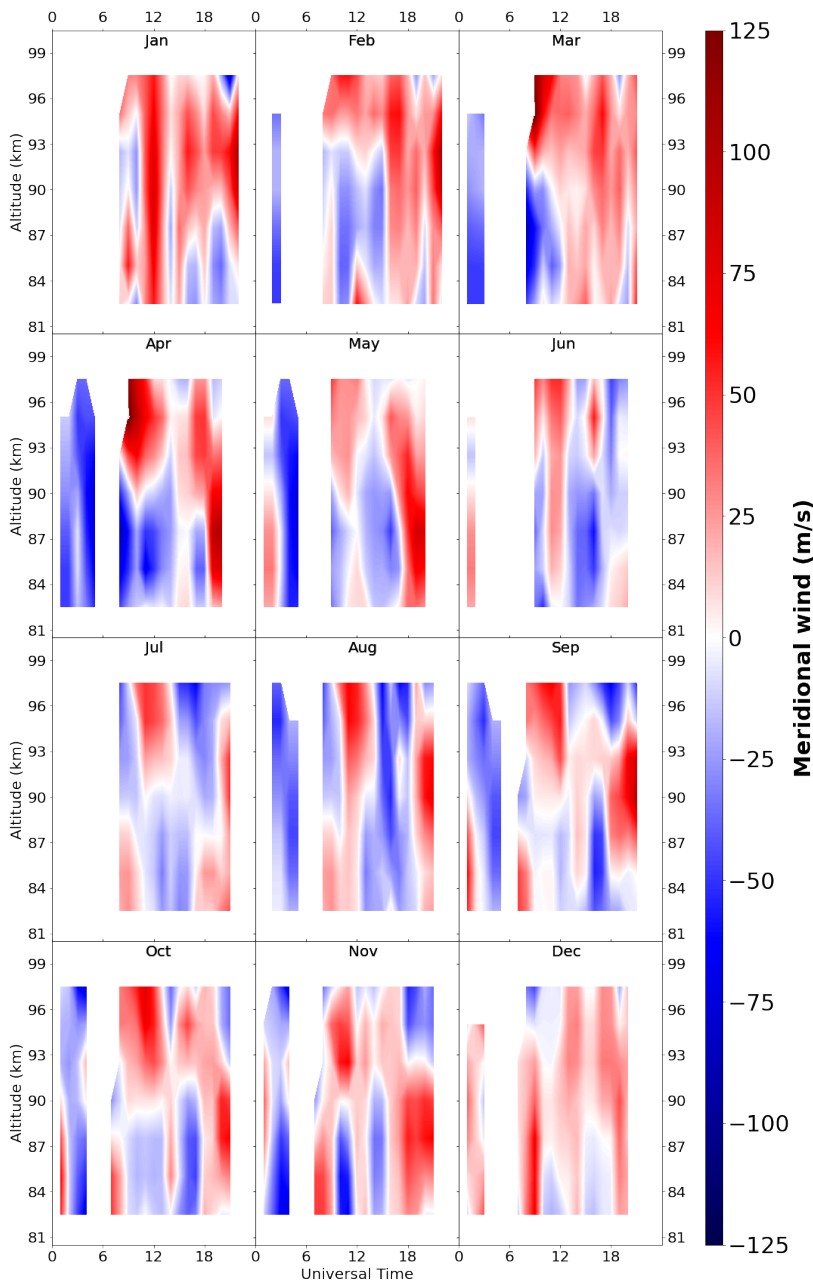

**Figure 3.** Same as Figure 2, but for the TIDI measurements.





In addition, some months presented the diurnal oscillation defined, however, short period structures are more evident for the TIDI meridional winds. Maybe the presence of the small oscillations along the day could mask the vertical propagation of the diurnal tide phases. The small number of soundings by the satellite within the chosen window could not be sufficient to average out the short time variation. John et al. (2011) compared the wind within temporal wind of three months and reached quite good agreements over Thumba ($8.5^o$ N, $77^o$ E), which is the equatorial region as well.

Figure 4 is the same as Figure 2, but for the zonal component. It is in the same scale as Figures 2, 3. Thus, one can observe that the zonal winds have small amplitudes than the meridional ones. Similarly, the diurnal oscillation is stronger during the summer, while the semidiurnal one appears sporadically for some altitudes.



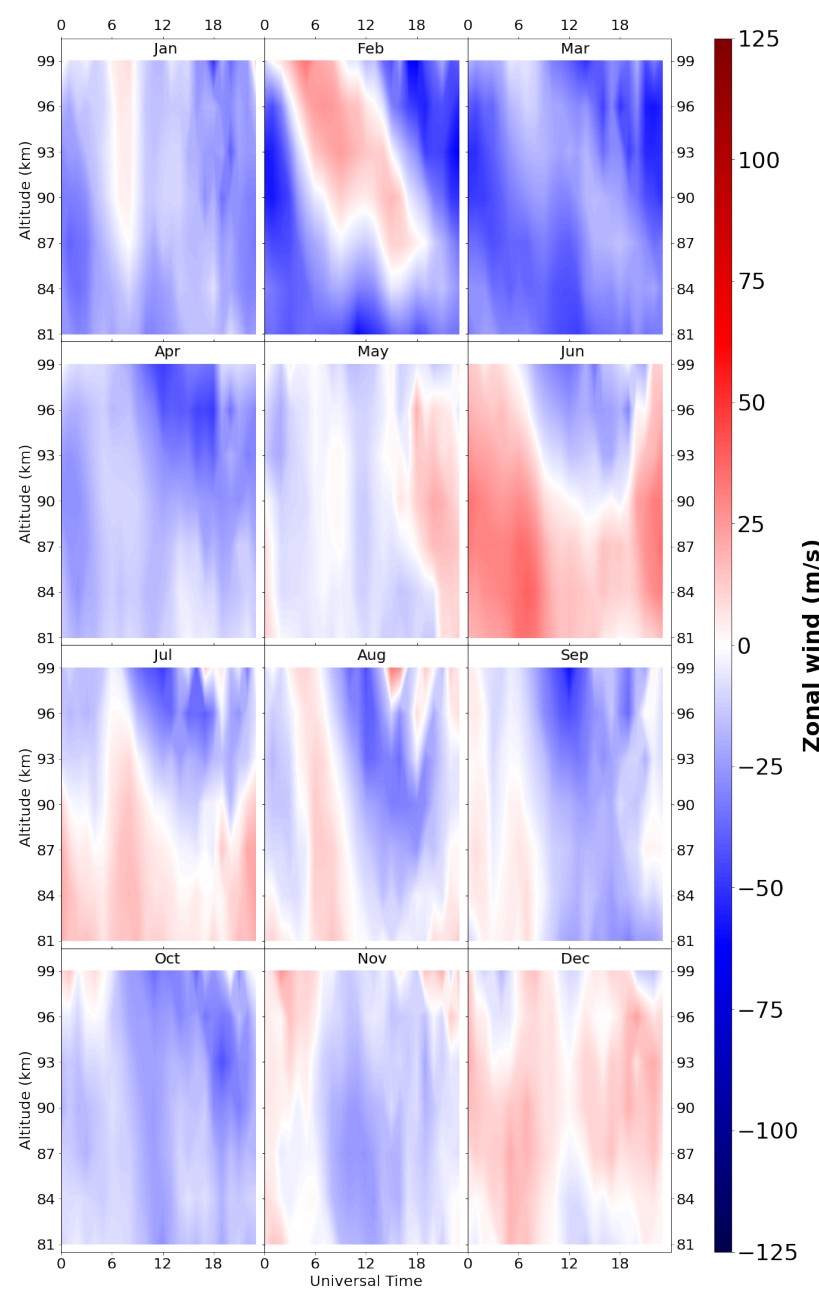

**Figure 4.** Monthly zonal mean wind calculated used the meteor radar for 2006.

Figure 5 presents the same kind of chart of Figure 3, but for the zonal component. For almost all months, the amplitudes of the zonal winds are larger than meteor radar ones. Diurnal structures are dominants but shorter periods structures appear as well for practically the whole year. Although, the mean zonal wind calculated from TIDI compares favorable to meteor radar





measurements, there are several short structures that could be associated with short period oscillation in the MLT as gravity waves, for instance. Comparisons from TIDI and Ionospheric Connection Explorer's (ICON's) Michelson Interferometer for Global High-resolution Thermospheric Imaging (MIGHTI) instruments showed consistency in the wind suggesting that the variations in the wind could be geophysical (Dhadly et al., 2021; Wu and Ridley, 2023). However, some variations could have
origin from the noise in the TIDI measurements as well.



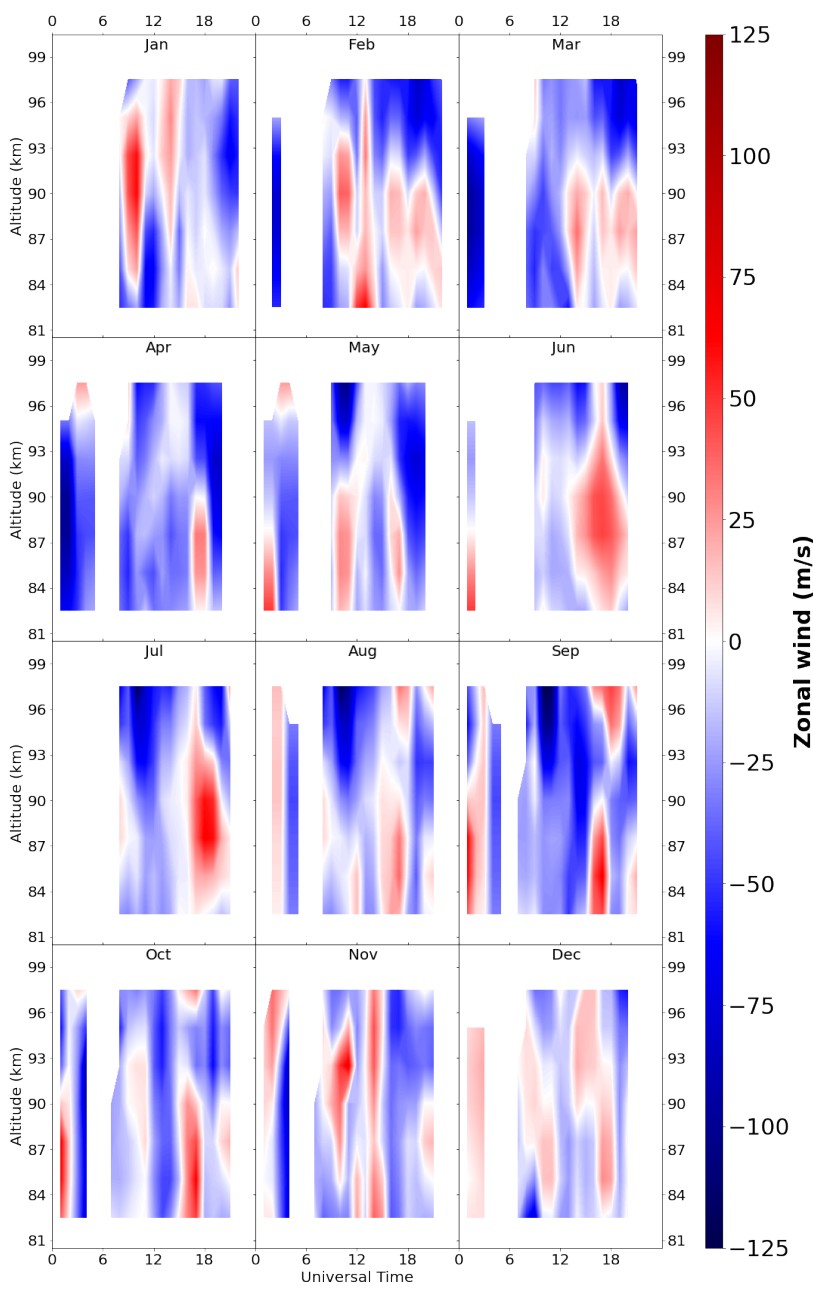

**Figure 5.** Same as Figure 4, but for the TIDI measurements.

Su et al. (2014) made a comparison of these two kinds of measurements during the Leonids meteor shower in 2012 and observed reasonable agreement, but short time structures were presented in the TIDI wind as well.





The last question to be discussed within the scope of these comparisons is how do the TIDI wind measurements respond to the seasonal, annual and semiannual variation. Features like quasi-biennial oscillation (QBO), semiannual oscillation and
annual oscillation have been pointed as responsible for the long term variability of the migrating diurnal tide (e.g., Xu et al., 2009).

Figure 6 shows the meridional wind measured by the two instruments along 2006 for 90 km altitude. The meteor radar wind points were taken for 12:00 UT while the TIDI winds were taken for the closest time in which the satellite crossed the window over São João do Cariri.

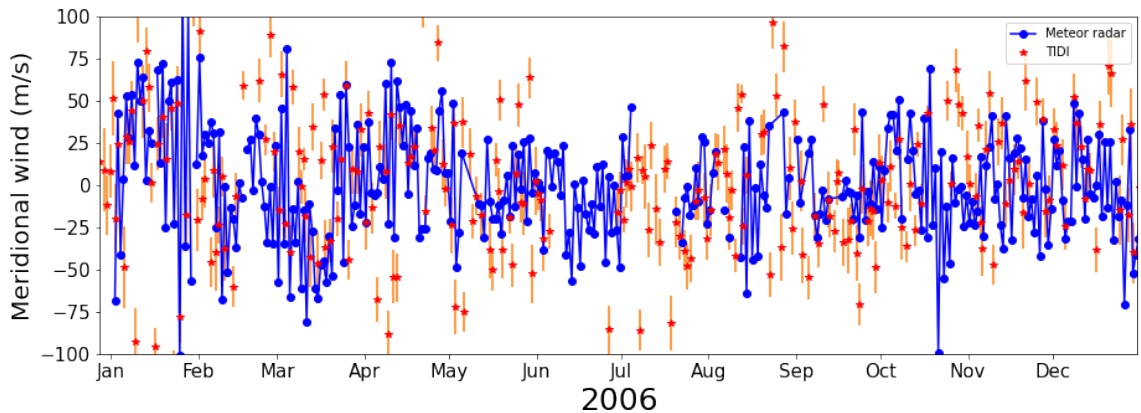

**Figure 6.** Temporal evolution of the meridional wind calculated at 90 km altitude for the meteor radar (blue) and TIDI (red) during 2006.

The meteor radar meridional wind presents an annual oscillation with maximum during the summer and an intraseasonal strong oscillation from January to May. Even the zonal wind from TIDI presenting spread points throughout the year, the points approach the general behaviour of the radar measurements.

Figure 7 is the same of Figure 6, but for the zonal component, which has a semiannual oscillation more pronounced and other short oscillation along the year. Again, the TIDI winds follows the meteor radar winds.

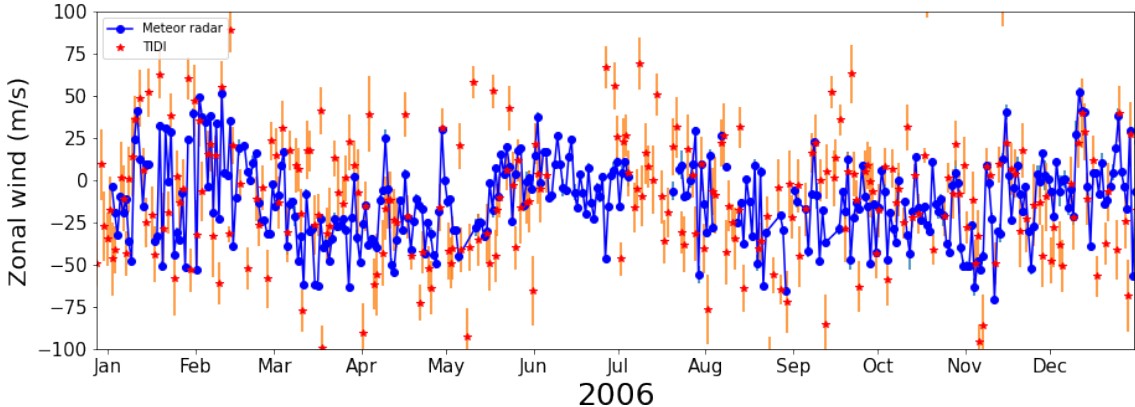

**Figure 7.** Same of Figure 6, but for the zonal component.

If one considers the measurements of Figure 6 and 7 obeying a statistical Gaussian distribution, Table 1 shows the average and standard deviation (SD) for the TIDI and meteor radar (MR) measurements. Note that the parameters of the Gaussian distribution are very close each other, except for the standard deviations that are greater for the TIDI measurements. Thereby, it suggest that the points of the two measurements, in addition to being close, they could obey the same statistical distribution.

**Table 1.** Statistical parameters for a Gaussian distribution for the zonal and meridional winds measured by the TIDI and meteor radar.

|  | Zonal average | Zonal SD | Meridional average | Meridional SD |
|---|---|---|---|---|
| **MR (m/s)** | -11.9 | 24.6 | -0.3 | 33.4 |
| **TIDI (m/s)** | -9.7 | 36.8 | 5.2 | 43.1 |

Xu et al. (2009) compared the amplitude of the migrating diurnal tide calculated from wind retrieved by these two technique
and the results showed good agreement as well. It suggests that for studies of long period observation, these measurements converges.

## 4   Conclusions

The present work compared the horizontal wind measured by the TIMED Doppler Interferometer and a meteor radar over São João do Cariri in 2006. Three aspects were analysed and discussed: (i) instantaneous measurements; (ii) daily behavior for
every month and (iii) the responses of the two techniques to the intraseasonal, semiannual and annual oscillations in the wind. The objective was to figure out advantages and disadvantages of each technique. So, the main conclusions are:

– Almost simultaneous measurements of the zonal and meridional wind vertical profiles could be substantially different comparing the TIDI and meteor radar measurements. It happens because the TIDI measures an instantaneous wind in the MLT region, while the meteor radar uses a bin size of one hour to average the wind over the whole sky. Thus, the TIDI



is more reliable to conduct studies involving short period waves (gravity waves) in the MLT. However, the disadvantage of using the TIDI to study gravity waves, for instance, is the difficult of matching simultaneous measurements from different instruments;

– Looking at the daily behaviour of the zonal and meridional winds calculated using the TIDI measurements for every month of 2006, there are qualitative agreements with the meteor wind calculations. However, the meteor radar calcu-

lations for each month is smoothly compared to the TIDI ones. For this reason, the meteor radar shows clearly the contribution of the tides (diurnal and semidiurnal) to the dynamics of the MLT. Extending the temporal window for integrating the daily wind from the TIDI measurements, the behaviours approaches each other;

– Both measurements respond satisfactorily to the long period (seasonal, semiannual and annual) oscillations and they could be comparable to studies of long term dynamics of the MLT.

*Data availability.* The meteor radar data can be requested to P.P. Batista (paulo.batista@inpe.br). TIDI data is available on line at https://timed.hao.ucar.edu/tidi/data.html

*Author contributions.* ARP - Conceptualization of this study, Methodology and Analysis; DOR and IP - Conceptualization, analysis and revision; LML - Conceptualization and revision; RAB, PPB and CW - Experiment and revision; AR - Experiment

*Competing interests.* I. Paulino is a member of the editorial board of ANGEO.

*Acknowledgements.* The present work has been supported by the Conselho Nacional de Desenvolvimento Científico e Tecnológico (#404971/2021-0, #306063/2020-4) and Fundação de Amparo à Pesquisa do Estado da Paraíba (grants #047/2021, 002/2019 e Edital Universal 09/2021)



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
