# Peer review of "Comparison of meteor radar and TIDI winds in the Brazilian equatorial region"

_Annales Geophysicae, 2023_

## Author Response (AR1)

**Responses to Editor and Reviewers**

**General Comments:**

Dear Dr. Gunter Stober.

Again, we appreciate for considering our manuscript suitable within the scope of Annales Geophysicae. We also thank the two reviewers for the comments and suggestions. Please, find enclosed the tracked changes file. Our point-by-point responses can be seen as follows.

Our apologies for the delay in submitting the revised version of the manuscript. As the reviewers suggested substantial revision, we needed to process the data again in order to recalculate the wind and re-write several statements of the manuscript. We consider very important and relevant the comments from both referees, which help us to improve the quality of the manuscript. After a careful revision and are resubmitting the manuscript for your appreciation.

**Referee #1:**

REFEREE: "This paper aims at investigating the horizontal (zonal and meridional) wind velocity at about 80-100 km altitude observed with the meteor radar at Sao Joao dos Cariri, Brazil and TIDI on board the TIMED satellite. I appreciate the devoted efforts by the authors to investigate characteristics of both ground-based radar and satellite measurements of the wind velocity in the mesosphere and lower thermosphere (MLT) region, where not so many effective wind velocity observations are available. However, I have major concerns regarding the statistical treatments on the comparison of the wind velocity data between the two techniques, as well as description and interpretation of the analyzed results"

AUTHORS: We appreciate the time that the Referee #1 dedicated to revise our manuscripts. In fact, the comments and suggestions were very interesting. We tried to do our best to address all concerns properly.

REFEREE: " Regarding (a), discrepancies between the meteor radar and TIDI are discussed, attributing to the difference in the field of view, the integration time of each measurement and the effects of gravity waves. However, I particularly do not agree that gravity waves significantly change the wave structure within four minutes, which are as short as the buoyancy period, but spatial separation between the two TIDI profiles may explain the discrepancy. Please show the horizontal extent of the meteor radar illuminating area and the locations of the TIDI measurements. In any case, single snap-shot is not enough to draw a conclusive remark. As the authors stated on Line 78, comparisons should be repeated for more cases."

AUTHORS: Yes, the Referee is absolutely right. We have revised those statements. Additionally, we have included Figure 1 (Figure 2 of the manuscript) which shows the field-of-view of the detected meteor echoes and the location of the TIDI soundings around 14:00-15:00 UT on 15 March 2006. One can observe that the horizontal distance between two TIDI's measurements is very large and it is likely the reason for the discrepancies between two consecutive measurements. Thank you for the suggestion!

[Figure]

Figure 1: Horizontal distribution of the meteor echoes detected on 15 March 2006 between 14:00 UT and 15:00 UT over São João do Cariri (red dots), which were used to compute the winds showed in Figure 1 of the manuscript. These meteor echoes were detected from ∼78 up to 102 km altitude. The red star shows the position of the TIDI's vertical soundings.

Figure 2 shows another example in which there are considerable discrepancies. We have tested several other coincident profiles, all of them do not match the TIDI and meteor radar profiles, which corroborate what we have sustained in the manuscript.

[Figure]

Figure 2: Vertical profiles of the meridional (blue) and zonal (red) winds measured by the meteor radar (solid lines) and TIDI (dashed, dot-dashed and dotted lines) over São João do Cariri. The meteor radar profiles were calculated at 10:00 UT on 01 June 2006. The TIDI measurements were retrieved at 10:54 UT (dashed lines) and 10:57 UT (dot-dashed lines).

REFEREE: **"For the comparison (b), it is explained as the 'daily mean win', but the figures seem to show the (local) time variations of the hourly wind velocity averaged over one month. Figure 2 with the meteor radar clearly shows the behavior of the diurnal and semidiurnal tides. On the other hand, Fig. 3 with TIDI includes irregular variations. Are the number of data comparable between the two determinations? "**

AUTHORS: Thank you for the suggestion to change the term "daily mean wind", we agree with the Referee and we have changed it in the manuscript. No, the number of measurements are not the same, For the meteor radar, we have a point per hour

almost every day, while from the TIDI can have more points in a single day, but for some days, we could not have points, because the size of the wind is small and it is possible that the satellite does not sounding within the window chosen. We have adjusted a threshold minimum of three measurements from the TIDI for considering the hourly mean wind. That is the reason for finding some blank spaces in Figures 3 and 4, which were updated in the manuscript as well.

[Figure]

Figure 3: Monthly time variation of averaged meridional wind calculated used the TIDI for 2006.

[Figure]

Figure 4: Same of Figure 3, but for the TIDI measurements.

REFEREE: **"I suspect Fig. 3 does not sufficiently smear out the waves other than tides, such as gravity waves, equatorial waves and so on, which are not synchronous to the local time. The same comments apply to Figs. 4 and 5."**

AUTHORS: Yes, we agree with the Referee and we have revised the text as well. Thank you!

REFEREE: **" The day-to-day variations of the wind velocity at 12 UT**

are shown in Figs. 6 and 7 at 90 km in year 2006. Because the one hour data at 12 UT is crucially affected by tides as recognized in Figs. 2 and 4, it does not represent the daily mean wind velocity. Instead, the wind velocity averaged throughout one full day should be used. Although the variations are explained in term of AO (annual oscillation), SAO (semiannual oscillation) and intra-seasonal oscillations, I do not clearly recognize signals of AO, SAO and others. At least, a harmonic analysis should be applied to identify AO and SAO. Statistical comparison over one full year, assuming the Gaussian distribution, may not be enough to show consistency between the two techniques. Day-to-day variations of the wind velocity and the decomposed AO and SAO signals should be statistically tested."

AUTHORS: Thank you for the suggestion. We have used the daily average for the meteor radar. In fact the presentation of the results is better now. We have also performed a least square fit as suggested by the Referee and it helps us to improve the quality of the Discussion. Figures 5 and 6 show those least square fits for each instruments.

[Figure]

Figure 5: Temporal evolution of the meridional wind calculated at 90 km altitude for the meteor radar (blue) and TIDI (red) during 2006. Solid blue line (meteor radar) and dashed red line (TIDI) represent the least square fits for AO, SAO and triannual oscillations (TAOs).

[Figure]

Figure 6: Same of Figure 5, but for the zonal component.

Additionally, we decide to keep the statistical comparison because it further corroborates the analysis. Figures shown below were incorporated to the manuscript as well.

REFEREE: " Overall, description and interpretation of the results are not fully convincing. Although the authors referred to earlier studies on related subjects, little discussions are given on agreement, discrepancy and progress compared to these published results."

AUTHORS: We agree with the Referee that the comparisons are poor, however, there are few publications on this topic to support our interpretations. Additionally, the focus of the present manuscript was quite different from previous publications, i.e., we are interested in comparing the advantages of each technique for the usage in the MLT investigation.

REFEREE: " I hope this study will be considerably improved, provided the authors revise the data analysis procedures and investigation of the results. However, I am sorry for not becoming positive to recommend publication of this manuscript in Annales Geophysicae."

AUTHORS: We agree with the main concerns from Referee # 1 and certainly, after the revision, the referee will be more comfortable to recommend our paper for publication. Thank you again for the valuable comments and suggestions.

REFEREE: " I am afraid I do not clearly understand some statements. I would like to recommend the manuscript be fully refined considering the specific comments listed below:"

AUTHORS: Thank you for the contribution reading carefully the statements and suggestion improvements. We have revised all statements according to the recommendations.

REFEREE: "L 9: 'vertical wind profiles' is misleading."

AUTHORS: It has been corrected.

REFEREE: **"L 16: Are 'the acoustic waves' recognized evident in the MLT region?"**

AUTHORS: We have removed this from text. Thank you

REFEREE: **"L 25-26: 'satellite measurement of wind' and 'using wind measurements' are redundant. In addition, 'wind' should be reworded as ´wind velocity' everywhere."**

AUTHORS: It has been fixed.

REFEREE: **"L 44: Is 'transceiver' commonly used to explain configuration of a radar?"**

AUTHORS: No, it is not. It's is more common use equipment or instrument. We have revised it.

REFEREE: **"L 48: Why is the meteor echo rate so variable between 1,000 and 3,000? I am also interested in the local time dependence of the meteor echo rate."**

AUTHORS: Depending on the position of the Earth in its orbit, the planet can find more or less particles, which introduce large variability of the detected echoes. During the day there is a strong variability as well, the radar detects more meteors in the early morning than the evening, it is explained by the motion in the orbit as well. For meteor radars deployed at middle and high latitude there is also a strong seasonal variation in the detection of echoes.

REFEREE: **"Description of a meteor radar system could be refined"**

AUTHORS: Thank you for the suggestion. We have added some details about the temporal dependency of the detection of echoes. A specification of the lower power radar was added to explain the number of the detected echoes per day. All relevant technical details of the radar are included as the power of operation, frequency and pulse rate, configuration of the transmitter and receiver antennas and physical principles used to detect the echoes and infer the wind.

REFEREE: **"Does 'mean wind' indicate the determination of the horizontal wind velocity in 4 km x 1 hour? How many meteor echoes in each bin?"**

AUTHORS: Yes, it does. The number of echoes used depending on the the day and the hour. There is a threshold minimum of 7 echoes to the determination of the wind.

REFEREE: **"L 53: Does 'vertical measurement' mean 'measurement of a vertical (height) profile'? Please refine the sentence."**

AUTHORS: In fact, we could not find this statement, but It does. We have revised the paragraph started in Line 53 of the old version of the manuscript.

REFEREE: **"Figure 1 and L 76-77, L79-80: Comparison of snap-shot profiles is not enough to derive the conclusive statements. If comparisons**

for more cases are expected to show a better agreement, please extend the analysis, and show the results."

AUTHORS: As we stated above in the main concern, it is very difficult to find good agreements between the measurements. Between the two sounds, it cold have large differences. Figure 1 shows another example. Furthermore, we have tested several other coincident profiles, all of them do not match the TIDI and meteor radar profiles, which corroborate with what we have sustained in the manuscript.

REFEREE: "L 80-84: I am curious why the ducting of gravity waves is explained in detail. Are their effects important in discussing the comparisons?"

AUTHORS: It is because the wind is an essential information for discussion on the formation of ducts. Several works have used MLT wind averaged within one hour bin. As the wind can change quickly, satellite measurements appear as an important tool.

REFEREE: "L 88: Why is TIDI advantageous to estimate Ri?"

AUTHORS: The reason is the same as written above. The meteor radar uses echoes from a large area in the sky and within the on hour interval. On the other hand, the satellite gives us an instantaneous wind.

REFEREE: "L 89-90: I simply do not understand this statement."

AUTHORS: We have revised it!

REFEREE: "L 92: 'climatological' is not an appropriate wording, as just one year data is analyzed."

AUTHORS: Yes, we agree with this. We have revised the word.

REFEREE: "Figure 2 and L 94: Fig. 2 does not seem to show 'daily mean winds', but the local time (diurnal) variations of hourly wind velocity averaged over one month."

AUTHORS: We agree with the referee and we have fixed it.

REFEREE: "Figure 2 and L 99: Because large amount of the meteor radar data is averaged, short period waves are smeared out, which do not synchronize with local time. Such averaging is not completely achieved for TIDI, so that irregular patterns appear in Fig. 3. Thus, the discrepancy is simply attributed to the amount of available data for TIDI, right? Please show the number of TIDI data used for comparisons."

AUTHORS: At least three measurements within an hour bin was necessary to calculate the mean wind. Please, note that there are blank spaces in the charts, which are results of this threshold.

REFEREE: "L 105-106: How does 'mask the vertical propagation of tides' mean? Are the small ooscillations (gravity waves) interacting with tides? Or, do they just visually overwrite the regular progression pattern

of tides? Why can the 'short period structures' be captured after averaging over 60 days, which do not seem to synchronize with local time?"

AUTHORS: We believe in the second option. It is not easy to verify. There are short period oscillations that can appear simply by the fact of having a small sample to average. We have revised this statement.

REFEREE: "L 108-109: I do not understand this statement. Is the accuracy of TIDI dependent of latitudes?"

AUTHORS: No it is not. We mean that the longer time interval (three months) could make the averages close to each other.

REFEREE: "Figures 4 and 5: The same concerns as in Figs. 2 and 3."

AUTHORS: We have also revised them.

REFEREE: "L 119-120: What is the 'noise' of TIDI?"

AUTHORS: Noise is associated with the error in the TIDI measurements.

REFEREE: "Figures 6 and 7: If the long-term variations, such as AO and SAO, are the target, daily mean wind is appropriate, rather than one hour data at 12 UT."

AUTHORS: Yes, we have revised it.

REFEREE: "Figure 6 and L 130: I do not clearly see AO with maximum in summer. How are the intra-seasonal oscillations detected?"

AUTHORS: As suggested, we have performed a least square fit to improve this discussion. Figures 5 and 6 shows those least square fits.

REFEREE: "L 131: 'even the zonal wind' Is the zonal wind shown in Fig. 6? Am I missing anything?"

AUTHORS: Yes, it is the meridional wind measured by the TIDE (red stars).

REFEREE: "Table 1: The Gaussian distribution over one full year is not very useful. Day-to-day variations of the wind velocity and the AO and SAO signals should be statistically tested."

AUTHORS: We agree with the referee. We decided to keep this analysis to corroborate, but, indeed, the least square fit is more useful in the discussion.

REFEREE: "'Section 4: Reviewing descriptions and explanations in Section 3, I am afraid I am not convinced with the concluding statements in this section.'

AUTHORS: I think that, after this revision, we are more confident in the statements.

REFEREE: "More appropriate references could be cited on earlier studies about the atmosphere dynamics and the measurement techniques.

**Some of self-citations to the author's group do not seem inevitable."**

AUTHORS: We have cited classical papers on the studies and the MLT as theoretical as experimental using meteor radar. In this version, we have also included some more citation like Jones et al. (1998) and Hocking and Thayaparan (1996) The "self citations" were used to help the reader to find more details about some specific topic that were suppressed in the manuscript to avoid redundancy. Even so, following the suggestion of the referee, we have removed some of them.

**Referee #2:**

REFEREE:"This manuscript presents a comparison between mesospheric obtained by a meteor radar located in north east Brazil (Sao Joao do Cariri), and the TIDI instrument onboard the TIMED satellite. The authors looked at almost coincident measurements, but also longer term observations. While the coincident measurements show a large variability and significant differences (even between 2 consecutive TIDI values), monthly averages and large-scale oscillations such as SAO, are in some kind of agreement. This paper is correctly written and but there is nothing really new at this stage. Similar comparisons have been done between radar wind measurements and satellite observations, as mentioned in the text (Xu et al., 2009, John et al., 2011, Su et al., 2014)."

AUTHORS: Thank you for your time revising our manuscript and for the important contributions suggested for our manuscripts. We agree with the Referee that there are previous works, which have done some kind of comparisons. For example, Xu et al. 2009 used meteor radar at low latitudes to corroborate their analysis, but the emphasis of that work was to study SAO and QBO. The paper by John et al., 2011, which used data from equatorial latitudes focused on the discussion of possible effects of the wind in the Equatorial Electrojet. They have compared the variation on the wind along the day averaging three months. Lastly, Su et al. (2014) investigated the wind during a meteor shower for a short time interval 10-25 November 2012 in middle latitudes. Based on it, we believe that the objective of our paper is completely different from the previous publication and released important contributions that pointed out advantages of each technique in the different kinds of the investigation in the MLT region. We have revised the manuscript considering all suggestions from Referee #2 in order to improve the quality of the presentation of the manuscript and the interpretations.

REFEREE:"Is it possible to have a map showing the coverage of both instruments (and at two times for TIDI), especially for Figure 1? There is a large discrepancy between two consecutive TIDI measurements, so it would be interesting to see if these measurements were made over mostly different regions."

AUTHORS: Yes, it is. Thank you for the suggestion we have included Figure 2, which shows the field-of-view of the detected meteor echoes and the location of the TIDI soundings around 14:00-15:00 UT on 15 March 2006. One can observe that the horizontal distance between two TIDI's measurements is very large and it is likely the reason for the discrepancies between two consecutive measurements.

REFEREE:"According to what you wrote (l. 72-74), because of the ways the two instruments work and what regions of the atmosphere they measure, it seems unlikely coincident measurements will be similar, so I'm not sure this part of the investigation is meaningful."

AUTHORS: Yes, we agree, however, the importance of this comparison is to show that the wind measured by the TIDI has a strong variability even between two consecutive measurements.

REFEREE:"l. 107: how many satellite soundings were used to plot the monthly mean winds? Did you use a threshold to limit the small sampling effect? Maybe you should."

AUTHORS: At least three measurements within an hour interval. Blank spaces represent intervals without measurements. We need to have in mind that the satellite crosses the São João do Cariri's area only one or two times in a day and the measurements are not sun synchronous.

REFEREE:"For Figure 6, you could have used the meteor data closest to the TIDI measurements instead of at 12 UT every day, especially considering the short-term variability in TIDI values, and the effects of tides on the winds."

AUTHORS: Thank you for the suggestion! Yes, we can do that or use a daily average as suggested by the Referee #1. In order to merge the suggestion of both referees, we have used daily average winds as shown in Figures 5 and 6, which were incorporated to the manuscript.

REFEREE:"l. 135: why should you consider that the measurements follow a Gaussian distribution? Instead, could you fit AO and SAO oscillations to both curves. It would be easier to compare them."

AUTHORS: Thank you for the suggestion, which is also in agreement with the suggestion of Referee #1. We have performed the least square fit considering AO, SAO and TAO. Indeed, it helped us during the discussion, thank you.

REFEREE:"l. 150: not sure TIDI is reliable to study short scale GW if there is so much variability in 3 min (Figure 1)."

AUTHORS: A single sample from TIDI is more reliable to study gravity waves because it is an instantaneous measurement. However, the distance between two soundings is large and it can provide distinct profiles. We have revised the statement in the manuscript.

REFEREE:"Minor edits... l. 5-6: ...which have a vertical resolution of 2.5 km, starting... '

AUTHORS: We have corrected it.

REFEREE:'l. 9-10: While TIDI... '

AUTHORS: We have corrected it.

REFEREE:'l. 13: for both techniques.'

AUTHORS: We have corrected it.

REFEREE:'l. 18: remove 'of the atmosphere'.'

AUTHORS: We have corrected it.

REFEREE:'l. 26: others. '

AUTHORS: We have corrected it.

REFEREE:' l. 29: oscillations.'
AUTHORS: We have corrected it.

REFEREE:'l. 32: hand, satellite measurements.'
AUTHORS: We have corrected it.

REFEREE:'l. 40: a meteor radar '
AUTHORS: We have corrected it.

REFEREE:'l. 47: to the ground. '
AUTHORS: We have corrected it.

REFEREE:'l. 54: wind components.'
AUTHORS: We have corrected it.

REFEREE:'l. 59: TIDI...using airglow emissions as...'
AUTHORS: We have corrected it.

REFEREE:'l. 60: airglow emissions. It has a... '
AUTHORS: We have corrected it.

REFEREE:'l. 62: there are vertical... '
AUTHORS: We have corrected it.

REFEREE:'l. 73: ... hand, TIDI... '
AUTHORS: We have corrected it.

REFEREE:'l. 83-84: remove 'in the MLT'.'
AUTHORS: We have corrected it.

REFEREE:'l. 85: their interaction with... '
AUTHORS: We have corrected it.

REFEREE:'l. 87: ratio . l. 88: TIDI. '
AUTHORS: We have corrected it.

REFEREE:'l. 89-90: can you clarify this sentence?'
AUTHORS: We have corrected it.

REFEREE:'l. 98: upper levels '
AUTHORS: We have corrected it.

REFEREE:'l. 101-102: TIDI. '

AUTHORS: We have corrected it.

REFEREE:'l. 102: within 60 days.'

AUTHORS: We have corrected it.

REFEREE:'l. 105: presented well-defined oscillations.'

AUTHORS: We have corrected it.

REFEREE:'l. 106-107: oscillations during the day could'modulate the observed diurnal tide phase. '

AUTHORS: We have corrected it.

REFEREE:'l. 111: smaller. '

AUTHORS: We have corrected it.

REFEREE:'l. 113: as Figure 3. '

AUTHORS: We have corrected it.

REFEREE:'l. 114: zonal winds are larger than the meteor radar ones...period. '

AUTHORS: We have corrected it.

REFEREE:'l. 115: favorably '

AUTHORS: We have corrected it.

REFEREE:'l. 119: remove 'in the wind'. '

AUTHORS: We have corrected it.

REFEREE:'l. 120: origined. '

AUTHORS: We have corrected it.

REFEREE:'l. 127: during 2006.'

AUTHORS: We have corrected it.

REFEREE:'l. 131: Even though... presented. '

AUTHORS: We have corrected it.

REFEREE:'l. 134: oscillations. '

AUTHORS: We have corrected it.

REFEREE:'. 135: Figures 6 and 7 obey... '

AUTHORS: We have corrected it.

REFEREE:'l. 137: close to each other. '

AUTHORS: We have corrected it.

REFEREE:'l. 138: the values of the two measurements... close, could obey... '

AUTHORS: We have corrected it.

REFEREE:'l. 148: TIDI. l. 155: smoother '

AUTHORS: We have corrected it.

REFEREE:'l. 159: studies long term dynamics in the MLT. '

AUTHORS: We have corrected it.

REFEREE:'Figure 1: switch the times (14:13 and 14:17) so they are chronological. Caption: ... winds measured... '

AUTHORS: We have corrected it.

REFEREE:'Figure 4: using'

AUTHORS: We have corrected it. We appreciate all minor suggestions by the referre #2

---

## Referee Report (RR1)

Comments on the paper "Comparison of meteor radar and TIDI winds in the Brazilian equatorial region" by Ana Roberta Paulino, Delis Otildes Rodrigues, Igo Paulino, Lourivaldo Mota Lima, Ricardo Arlen Buriti, Paulo Prado Batista, Aaron Ridley, and Chen Wu.

The theme of this study is relevant for the journal. However, the analysis is poorly described and is not accurate. The study needs additional data processing and more comprehensive analysis.

Detail comments

Abstract. The authors write in the abstract that they use a grid of -10 - +10 degrees. However, the reader find in the text, that a grid of -20 - +20 degrees was used.

1. The first question that naturally arises: why 2006 year, why only 2006?
2. According to the rules for the TIDI data analysis the authors should clearly indicate the data type and the data version used for the analysis. "It is recommended that TIDI data users specify these version numbers when publishing results to avoid any uncertainty related to the origin of the data."
3. The authors do not provide a detail description of the TIDI data processing. It is not clear what time interval they use to estimate the TIDI mean winds and how they estimate the winds. The correct procedure employs at least a 60-day time interval. Even the 60-day time interval is not always enough. A few gaps in the local time coverage could be obtained. It is not clear: how the authors deal with gaps, how the authors deal with seasonal changes and long-period variations.
4. It is not clear why the authors presented fig.1. The TIDI instantaneous profile variability is well known (see, references in the manuscript). The comparison is doubtful as described by the authors.
5. Page 5. Incorrect reference to John et al. (2011). They used much longer time interval to calculate the wind profiles.
6. Fig.7 The authors use the fitting of the meteor hourly mean winds but the separate TIDI profile data. This approach does not take into account that the TIDI data may provide many profiles for some local hours and significantly fewer for the others.
7. The authors write: "**Figure** 7 and 8 **obey** a statistical Gaussian distribution". This is an incorrect statement. Please, change.

8. Conclusions

Ln. 150. The authors draw very general conclusions based on a couple of examples analyzed in the work. It is even impossible to say about any statistical analysis. Therefore, I propose to remove this and the next one conclusion from the text.

Ln. 170. The authors state, that: "Extending the temporal window for integrating the daily wind from the TIDI measurements, the behaviours approaches each other"
Sorry, I didn't find this type of an analysis in the text.

Table 1. The authors state that the TIDI wind data obey the Gaussian distribution. Please, provide statistical arguments for this statement. In fact, it is not necessary to have the Gaussian distribution to find the mean and standard deviation.

---

## Referee Report (RR2)

Comments on the paper "Comparison of meteor radar and TIDI winds in the Brazilian equatorial region" by Ana Roberta Paulino, Delis Otildes Rodrigues, Igo Paulino, Lourivaldo Mota Lima, Ricardo Arlen Buriti, Paulo Prado Batista, Aaron Ridley, and Chen Wu.

The theme of the study is relevant to the journal. However, I believe that results of the data analysis can be significantly more informative, and the study requires additional efforts. I think it will not take a lot of time.

Detail comments
Abstract. The authors write in the abstract that they use a grid of ±5 degrees. However, the reader may find in the Introduction, that a grid of ±10 was used.

1. Fig.5-6. The comparison shows a large difference between MR and TIDI winds. Therefore, a question arises about the method of the comparison. I would like to repeat a part of my previous comments about the TIDI data processing.
It is unclear: how the authors deal with gaps, how the authors deal with wind seasonal changes and long-term wind oscillations. Wind speeds at different LT hours were taken from different days. Therefore, planetary waves or strong prevailing wind changing will create additional short-term variability. The MR data allow to check this effect, the MR wind can be taken at LT hours of the TIDI winds.
Also, there is a limit for large MR winds to remove unphysically large values. Did the authors use any limit values for the TIDI winds?
Perhaps it is better to provide a comparison between prevailing winds, diurnal and semidiurnal tides.
2. Ln 130. Unclear statement: "Maybe the presence of the small oscillations oscillations during some days could modulate the observed diurnal tide phase. " Why can't large oscillations modulate the tidal phase?

3. Fig. 1 Indeed, the TIDI wind profiles are not instantaneous. There is need of about 2 minutes to obtain LOS wind velocity.

4. Fig 7-8. Please, show the seasonal wind changes for all available heights.

Table 1. Fig.7-8 show the seasonal wind changes, but the reader can find the parameters in Table 1 for the whole year 2006. It seems reasonable to present the parameters for different seasons.

Summary.
The authors state that "there are qualitative agreements with the meteor wind calculations. However, the meteor radar calculations for each month is smoother compared to the TIDI ones". The agreement seems to be much worse. This is also an important result. In light of my comment 1, I propose a different formulation.

---

## Author Response (AR2)

**Responses to the Editor and Referees**

**General Comments:**

Dear Dr. Gunter Stober.

EDITOR: **"today, I received the second round of reviewer assessment. Both reviewers were newly assigned. Judging from the reports, the manuscript requires major revision. In particular, the points raised in report #2 should be considered and the manuscript should be revised carefully including these comments.The revised manuscript will be assessed by both reviewers and the editor."**

AUTHORS: Thank you for conducting the revision process of our manuscript. We also express our gratitude to the valuable comments from the referees, which help us to improve this manuscript. We did our best to properly address all concerns from them.

Please, find below our point-by-point responses to the referees and we have also tracked changes in the manuscript to make easy the third round of revision by the Referees.

Best regards,

The authors.

**Referee #1:**

REFEREE:"I was asked to review the revised manuscript entitled 'Comparison of meteor radar and TIDI winds in the Brazilian equatorial region'. This manuscript focuses on the comparison of horizontal neutral winds obtained from a meteor radar, and the TIDI on board the TIMED satellite. It was found that substantial differences existed between the measurements from the two types of instruments. This difference, however, is expected to exist because that the wind from meteor radar is space-time-averaged wind and the wind from TIDI is almost instantaneous wind. It's difficult to get the new/important points of this manuscript in its present form, whereas such a comparison of winds by the two types of instruments may be important and valuable."

AUTHORS: We really appreciate the kindly acceptance of the Referee #1 to revise the second version of the manuscript. The referee has pointed out important concerns and we have done our best to address them. Regarding the importance of this manuscript, although the two techniques for measuring the wind are different, we have worked in this manuscript to show to the readers which technique is more appropriate to be used depending on the time scale of the investigation. In this aspect, the manuscript offers an important contribution, primarily to the community that does not have expertise in wind measurements in the MLT, but needs this kind of data to advance in different studies.

REFEREE: "Page 3, line 73, please briefly mention why consider the grid ±10 degree of latitude and longitude around the meteor radar. Is it because that the spatial coverage of meteors detected by the radar covers ±10 degree? (lines 4, 41, 71-73, 122, of the manuscript file)."

AUTHORS: Thank you for this comment. We have changed the size of the window to match with the field of view of the meteor radar, which is plus or minus 5 degrees. We have changed the text as suggested.

REFEREE: "Figure 2. Please mark the location of meteor radar in Figure 2. For meteor backscatter observation, less echoes are usually expected at the locations right over the radar site (-7.4S, -36.5W). 'red star' should be 'blue star'. "

AUTHORS: Thank you for the suggestion. We have marked the point where the radar is deployed as a black triangle. We have also corrected the text in the manuscript (Caption of Figure 1). Please see in Figure 1 (of this document) the integrated distribution of the detected echoes for 15 March 2006, which is in agreement with the Referee comment.

[Figure]

Figure 1: Horizontal distribution of the meteor echoes detected on 15 March 2006 over São João do Cariri (red dots). These meteor echoes were detected from ∼78 up to 102 km altitude.

REFEREE: **"Figure 8. The zonal wind derived from meteor radar shows a semiannual oscillation, but the zonal wind from TIDI shows a triannual oscillation with three peaks around March, July and November. Please discuss the difference."**

AUTHORS: Thank you for this observation. Indeed the triannual component is stronger in the satellite data, but we are not sure if it is a real behaviour of the wind or it comes from the spreading of the data. For example, Figure 2 shows the zonal wind for a grid of the same size centered at (27ºS, 6ºW), arbitrarily chosen. It can be seen that the behavior is quite different.

[Figure]

Figure 2: Zonal daily wind calculated using the TIDI measurement for 90 km at (27ºS, 6ºW).Dashed line show a least square fit for annual, semiannual e triennial oscillation.

Additionally, Figure 3 shows many spreading points from the TIDI that can contribute to this behavior. So, following the suggestion of the Referee #1, we mention the presence of this oscillation in the manuscript (lines 154-155), but we prefer to hold on to a deep discussion for a while, because it is out of the scope of the manuscript.

[Figure]

Figure 3: Same of Figure 2, but for São João do Cariri and including the meteor radar data (blue symbols).

**Referee #2:**

REFEREE:"The theme of this study is relevant for the journal. However, the analysis is poorly described and is not accurate. The study needs additional data processing and more comprehensive analysis. "

AUTHORS: We thank the important comments of the Referee #2, which kindly agree in revise our manuscript and contribute to improve it. We have followed the suggestions in order to address what was pointed out by the Referee #2.

REFEREE:"Abstract. The authors write in the abstract that they use a grid of -10 - +10 degrees. However, the reader find in the text, that a grid of -20 - +20 degrees was used."

AUTHORS: Thank you for checking this mistake. We have revised the text and corrected it following the suggestion of the Referee #1 (lines 4, 41, 71-73, 122, of the manuscript file).

REFEREE:"The first question that naturally arises: why 2006 year, why only 2006?"

AUTHORS: Thank you for asking. 2006 was the first year of operation of the São João do Cariri's meteor radar. During this year, we obtained very good quality data. Figure 7 and 8 of the manuscripts shows the complete sequence of the measurements (without gaps for this year). We have included this motivation in this manuscript (lines 73-75). Furthermore, for this kind of study, which the main objective is to compare the advantages of each technique, one year is enough.

REFEREE:"According to the rules for the TIDI data analysis the authors should clearly indicate the data type and the data version used for the analysis. It is recommended that TIDI data users specify these version numbers when publishing results to avoid any uncertainty related to the origin of the data."

AUTHORS: Thank you for the comment. We have added the description of the data type and version in the section 'Data availability'.

REFEREE:"The authors do not provide a detail description of the TIDI data processing. It is not clear what time interval they use to estimate the TIDI mean winds and how they estimate the winds. The correct procedure employs at least a 60-day time interval. Even the 60-day time interval is not always enough. A few gaps in the local time coverage could be obtained. It is not clear: how the authors deal with gaps, how the authors deal with seasonal changes and long-period variations."

AUTHORS: The referee is right regarding the presence of gaps. We have used a 60-day wind to compute the composite days in Figures 4 and 6 of the manuscript. One can see that there are gaps in Figures 4 and 6. If we enlarge the window size, those gaps diminish (One can see it in Figures 4 and 5 of this document). However, we are using a window size comparable to the field of view of the meteor radar (it has been requested by the Referee #1). On the other hand, the objective of the manuscript is to compare the advantages of the two techniques. Then, the presence

of the gaps help us to discuss that for the usage of the TIDI data in climatological winds, it is necessary a wind size larger than the field of view of the meteor radar and time interval as long as 60 days. We have added a statement explaining the question of the Referee #2 about the temporal and spatial window size (lines 121-127). Additionally, we have also commented about the seasonal changes in the manuscript. Thank you for this important comment.

REFEREE:**"It is not clear why the authors presented fig.1. The TIDI instantaneous profile variability is well known (see, references in the manuscript). The comparison is doubtful as described by the authors."**

AUTHORS: Thanks for this comment.Yes, the Referee is right and this concern was discussed in the manuscript. The objective was to compare this well known variability with the meteor radar mean profile. We have pointed out that depending on the objective of the study, the measurements from the TIDI has advantage because they show the real and instantaneous condition of the atmosphere. We have added this explanation in the manuscript to address the concern of the Referee #1 (lines 80-82).

REFEREE:**"Page 5. Incorrect reference to John et al. (2011). They used much longer time interval to calculate the wind profiles."**

AUTHORS: Our apologies for this mistake, we have corrected it (lines 89-90).

REFEREE:**"Fig.7 The authors use the fitting of the meteor hourly mean winds but the separate TIDI profile data. This approach does not take into account that the TIDI data may provide many profiles for some local hours and significantly fewer for the others."**

AUTHORS: The referee is right. We have corrected it and used the same methodology, i.e., we have used one day bin to average the TIDI data as well. It was corrected in the text. (lines 246-147)

REFEREE:**"The authors write: 'Figure 7 and 8 obey a statistical Gaussian distribution'. This is an incorrect statement. Please, change."**

AUTHORS: Thank you for this observation. We have correct it (lines 155-156).

REFEREE:**"Conclusions Ln. 150. The authors draw very general conclusions based on a couple of examples analyzed in the work. It is even impossible to say about any statistical analysis. Therefore, I propose to remove this and the next one conclusion from the text."**

AUTHORS: Thank you for the suggestions.We have revised them including the information that we have used case studies in the present manuscript (line 16)). Additionally, we have changed the section name to "Summary", which, certainly will address the concerns of the Referee #2.

REFEREE:**"Conclusions Ln. 170. The authors state, that: 'Extending the temporal window for integrating the daily wind from the TIDI mea-**

**surements, the behaviours approaches each other.' Sorry, I didn't find this type of an analysis in the text."**

AUTHORS: The Referee is right. We have suppressed these Figures from the manuscript because we think they are not necessary. We have now revised the text of the manuscript to emphasises this (lines 125-128, 178-179). Figures 4 and 5 of this document shows the variation of the mean wind along the day for the TIDI measurements enlarging the size of the window to 30 x 30 degrees. Please note that the gaps were reduced compared to Figure 4 and 6 of the manuscript.

[Figure]

Figure 4: Monthly time variation of averaged meridional wind for 2006 calculated used the TIDI for a window of $30 \times 30$ degrees latitude x longitude centered at São João do Cariri for 2006 with a temporal window of 60 days.

[Figure]

Figure 5: Same of Figure 4, but for the zonal component.

REFEREE:"**Table 1. The authors state that the TIDI wind data obey the Gaussian distribution. Please, provide statistical arguments for this statement. In fact, it is not necessary to have the Gaussian distribution to find the mean and standard deviation.**"

AUTHORS: Thank you for this important comment. We agree with the Referee

**2 that it is not necessary that the data obey a gaussian distribution to calculate the mean and standard deviation. Fortunately, in this case they obey as shown in Figures 6 and 7 of this document. We have revised the text according to the Referee suggestion (line 156).**

[Figure]

Figure 6: Histogram for the meridional component of the wind measured by the meteor radar (blue) and TIDI (red). Gaussian fit were over plotted to both histograms.

[Figure]

Figure 7: Same of Figure 6, but for the zonal component.

---

## Author Response (AR4)

**Responses to the Editor and Referees**

**General Comments:**

Dear Dr. Gunter Stober.

EDITOR: "**one reviewer is now pleased, but the other reviewer still has major concerns. I did read the comments and believe it is easy to implement them. The raised points are reasonable and it is recommended to follow the suggestions. Please submit a revised version of your manuscript highlighting the changes according to the reviewer's suggestion.** "

AUTHORS: Thank you for your support. We have revised the manuscript considering the comments from the Referee #2. We have marked the changes in the tracked changes file and our point-by-point reply is following this letter.

Best regards,

The authors.

**Referee #1:**

REFEREE: **"accepted as is"**

AUTHORS: We appreciate the contribution from the Referee #1 and the recognition that was given for our efforts to address the concerns pointed out.

**Referee #2:**

REFEREE: "Comments on the paper 'Comparison of meteor radar and TIDI winds in the Brazilian equatorial region1 by Ana Roberta Paulino, Delis Otildes Rodrigues, Igo Paulino, Lourivaldo Mota Lima, Ricardo Arlen Buriti, Paulo Prado Batista, Aaron Ridley, and Chen Wu. The theme of the study is relevant to the journal. However, I believe that results of the data analysis can be significantly more informative, and the study requires additional efforts. I think it will not take a lot of time."

AUTHORS: We appreciate the time of the Referee #2 revising the manuscript and we thank for the important suggestions, which certainly will improve the quality of the manuscript. We did our best to address all unclear points as suggested by the Referee #2.

REFEREE: "Abstract. The authors write in the abstract that they use a grid of ±5 degrees. However, the reader may find in the Introduction, that a grid of ±10 was used."

AUTHORS: Thank you for this important observation. Initially, we have worked with a ±10 degrees box, however, following several suggestions during the peer review process, we have changed the analysis to a box of ±5 degrees. We have changed it in the Introduction.

REFEREE: "Fig.5-6. The comparison shows a large difference between MR and TIDI winds. Therefore, a question arises about the method of the comparison. I would like to repeat a part of my previous comments about the TIDI data processing. It is unclear: how the authors deal with gaps, how the authors deal with wind seasonal changes and long-term wind oscillations. Wind speeds at different LT hours were taken from different days. Therefore, planetary waves or strong prevailing wind changing will create additional short-term variability. The MR data allow to check this effect, the MR wind can be taken at LT hours of the TIDI winds."

AUTHORS: Thank you so much for this important comment. We agree with the total concern of the referee and it was one of our concerns as well. Due to the quasi-sun synchronous orbit of the TIMED, a time interval of about 60 day is necessary to cover an entire day. Even using a 60 day window, we can observe several gaps in Figure 4 and 6. Extending this window, it is likely that the gaps will be reduced, however, the seasonal changes, as pointed out by the referee, can change the pattern of the winds. So, after several tests, we decided to keep a 60 day window, even with the presence of the gaps. In fact, it is an important result for the proposal of the manuscript that shows the limitation of this instrument to conduct this kind of study with TIDI data. As the meteor radar has continuous measurement every day, Figures 3 and 5 do not present gaps and the values are smoothed compared to the TIDI ones. The basic idea of this comparison was to check whether within a reasonable time window, the TIDI could reproduce the same behavior of the meteor winds. It is important to remember that in Figure 1, we compared almost simultaneous measurements and the profiles were quite different from each other. In this case, we have averaged all TIDI wind profiles within a time range along the

two months as a representative month.

REFEREE:**Also, there is a limit for large MR winds to remove unphysically large values. Did the authors use any limit values for the TIDI winds? Perhaps it is better to provide a comparison between prevailing winds, diurnal and semidiurnal tides.**"

AUTHORS: Thank you for this question and comment. We have used the same filter of the meteor winds in the TIDI winds, i.e., winds faster than 150 m/s were considered missed points. Comparing prevailing wind and tides is indeed a good suggestion for future work, thank you for suggesting this. However, we understand this kind of analysis is out of the scope of the present manuscript. We would have to implement, test and validate a methodology to calculate tides from the TIDI data and certainly it will take a long time to be done.

REFEREE:**"Ln 130. Unclear statement: 'Maybe the presence of the small oscillations oscillations during some days could modulate the observed diurnal tide phase.' Why can't large oscillations modulate the tidal phase? "**

AUTHORS: The referee is right. As small as large oscillation could modulate the tidal phase. We have fixed the statement. Thank you for the suggestion.

REFEREE:**"Fig. 1 Indeed, the TIDI wind profiles are not instantaneous. There is need of about 2 minutes to obtain LOS wind velocity."**

AUTHORS: Yes, the referee is right. We have changed it to "quasi instantaneous" and explained it in the Introduction. Thank you.

REFEREE:**"Fig 7-8. Please, show the seasonal wind changes for all available heights."**

AUTHORS: Thank you for the suggestion. We have changed Figure 7 and 8 according to the Referee suggestion. In fact, it seems to be better because it reduces the spread points. There are no significant differences regarding what is explored in these figures. We have also tested other individual altitudes and, in general, the behavior is similar. These figure are show as following:

[Figure]

Figure 1: Temporal evolution of the meridional wind calculated for all available altitudes for the meteor radar (blue) and TIDI (red) during 2006. Solid blue line (meteor radar) and dashed red line (TIDI) represent the least square fits for AO, SAO and triannual oscillations (TAOs).

[Figure]

Figure 2: Same of Figure 1, but for the zonal component.

REFEREE:**"Table 1. Fig.7-8 show the seasonal wind changes, but the reader can find the parameters in Table 1 for the whole year 2006. It seems reasonable to present the parameters for different seasons."**

AUTHORS: Thank you for the suggestion, we have added some rows showing the parameter for all seasons as can see in Table 1 . It was also incorporated to the manuscript.

Table 1: Statistical parameters for a Gaussian distribution for the zonal and meridional winds measure by the TIDI and meteor radar.

|          |            | Zonal AVG | Zonal SD | Merid. AVG | Merid. SD |
|----------|------------|-----------|----------|------------|-----------|
| Total    | **MR (m/s)**   | -8.9  | 18.6 | -1.0 | 10.2 |
|          | **TIDI (m/s)** | -14.3 | 23.0 | -0.4 | 21.8 |
| Summer   | **MR (m/s)**   | -17.7 | 18.0 | 1.0  | 13.7 |
|          | **TIDI (m/s)** | -13.0 | 24.0 | 2.2  | 24.6 |
| Fall     | **MR (m/s)**   | -7.3  | 18.9 | -2.5 | 8.5  |
|          | **TIDI (m/s)** | -16.8 | 18.5 | 4.1  | 24.1 |
| Winter   | **MR (m/s)**   | -2.7  | 14.8 | -5.6 | 7.0  |
|          | **TIDI (m/s)** | -16.3 | 27.3 | -9.7 | 18.0 |
| Springer | **MR (m/s)**   | -6.8  | 18.5 | 1.7  | 8.4  |
|          | **TIDI (m/s)** | -11.9 | 20.7 | 2.3  | 17.5 |

REFEREE:"**Summary.The authors state that 'there are qualitative agreements with the meteor wind calculations. However, the meteor radar calculations for each month is smoother compared to the TIDI ones'. The agreement seems to be much worse. This is also an important result. In light of my comment 1, I propose a different formulation..**"

AUTHORS: Yes, the referee is right. We have reformulated this statement. Thank you for the suggestion.